# Comparative Analysis of Policosanols Related to Growth Times from the Seedlings of Various Korean Oat (*Avena sativa* L.) Cultivars and Screening for Adenosine 5′-Monophosphate-Activated Protein Kinase (AMPK) Activation

**DOI:** 10.3390/plants11141844

**Published:** 2022-07-14

**Authors:** Han-Gyeol Lee, So-Yeun Woo, Hyung-Jae Ahn, Ji-Yeong Yang, Mi-Ja Lee, Hyun-Young Kim, Seung-Yeob Song, Jin-Hwan Lee, Woo-Duck Seo

**Affiliations:** 1Crop Foundation Research Division, National Institute of Crop Science, Rural Development Administration, Jeollabuk-do, Wanju-gun 55365, Korea; gajae93@gmail.com (H.-G.L.); hengja112@gmail.com (H.-J.A.); yjy90@korea.kr (J.-Y.Y.); esilvia@korea.kr (M.-J.L.); hykim84@korea.kr (H.-Y.K.); s2y337@korea.kr (S.-Y.S.); 2Division of Life Sciences, College of Natural Science, Jeonbuk National University, 567 Baekje-daero, Jeollabuk-do, Jeonju 54896, Korea; 3Natural Medicine Research Center, Korea Research Institute of Bioscience and Biotechnology, Chungcheongbuk-do, Cheongju-si 28116, Korea; woosy@kribb.re.kr; 4Department of Agbiotechnology and Natural Resources, College of Agriculture and Life Science, Gyeongsang National University, Jinju 52828, Korea; 5Department of Life Resources Industry, College of Natural Resources and Life Science, Dong-A University, 37, Nakdong-daero 550 beon-gil, Busan 49315, Korea

**Keywords:** oat seedling, policosanol, hexacosanol, AMPK, growth times, GC-MS

## Abstract

The objectives of this research were to evaluate the policosanol profiles and adenosine-5′-monophosphate-activated protein kinase (AMPK) properties in the seedlings of Korean oat (*Avena sativa* L.) cultivars at different growth times. Nine policosanols in the silylated hexane extracts were detected using GC-MS and their contents showed considerable differences; specifically, hexacosanol (6) exhibited the highest composition, constituting 88–91% of the total average content. Moreover, the average hexacosanol (6) contents showed remarkable variations of 337.8 (5 days) → 416.8 (7 days) → 458.9 (9 days) → 490.0 (11 days) → 479.2 (13 days) → 427.0 mg/100 g (15 days). The seedlings collected at 11 days showed the highest average policosanol content (541.7 mg/100 g), with the lowest content being 383.4 mg/100 g after 5 days. Interestingly, policosanols from oat seedlings grown for 11 days induced the most prevalent phenotype of AMPK activation in HepG2 cells, indicating that policosanols are an excellent AMPK activator.

## 1. Introduction

Policosanols are long-chain aliphatic primary alcohols that were first extracted from sugar cane (*Saccharum officinarum* L.) at Dalmer Laboratories in Cuba and mainly composed of policosanol, octacosanol (C-28), triacontanol (C-30), and hexacosanol (C-26) [1,2]. These metabolites were also contained minor compositions including tetracosanol (C-24), heptacosanol (C-27), nonacosanol (C-29), dotriacontanol (C-32), and tetratriacontanol (C-34) [1]. Moreover, policosanol monotherapy reduces low-density lipoprotein (LDL)-cholesterol levels by increasing the expression of LDL receptors and increases high-density lipoprotein (HDL)-cholesterol levels [3,4]. Policosanol has recently shown several pharmacological activities such as reduction of platelet aggregation and antiulcer, antioxidant, and anti-inflammatory activities [1,5,6,7,8]. Specifically, these components and their natural sources have increased interest in the food and medical industries for use in emulsion, capsule, and tablet delivery systems, owing to their bioavailability and efficacy [1]. It is well established that the policosanols in crops shows significant differences according to various factors including environmental conditions and genetics, as reported in the literatures [9,10,11,12]. Moreover, many researchers have reported that the wheat grains contain policosanols and phytosterols, and their contents as well as compositions are positively correlated with the growth conditions and management [9,13,14]. Recently, several studies have demonstrated that natural plants and crops are adenosine-5′-monophosphate-activated protein kinase (AMPK) activators and the policosanol content plays an important role in determining the rate of AMPK activation [12,15]. Notably, AMPK regulates energy metabolism such as glucose and cholesterol as well as hepatic lipids, and AMPK activation blocks ATP-consuming anabolic pathways, including fatty acid, cholesterol, and protein syntheses [12]. Moreover, Policosanol is known to regulate AMPK-mediated cholesterol synthesis reduction and 3-hydroxy-glutaryl-CoA reductase (HMGCR) reductase activity in HepG2 cells [16,17]. In recent, the extracts of barley and wheat seedlings containing high levels of policosanols can reduce plasma cholesterol concentrations via the activity of the AMPK-dependent phosphorylation inhibition-limiting enzyme in cholesterol biosynthesis, HMGCR. Especially, hexacosanol (C26-OH), a major policosanol in barley seedlings, displayed considerable AMPK activation abilities [15].

Among diverse crops, oat (*Avena sativa* L.), belonging to the family *Gramineae*, is one of the most popular healthy foods worldwide and contains several biological metabolites, including carbohydrates, sterols, lipids, proteins, alkaloids, saponins, and flavonoids. This crop is of great important in the prevention and treatment of diseases in herbal remedies because it contains esters, phospholipids, triglycerides, and fatty acids [18,19]. For these above reasons, several researchers have recently documented the various biological properties such as antioxidant, antimicrobial, antidiabetic, anti-inflammatory, antiplatelet, and antiparasitic effects [20,21]. In addition, we have recently reported that the OSs exhibited anti-osteoporotic activity [22]. However, to the best of our knowledge, no studies have performed comparative analyses of policosanol derivatives and biological abilities in various cultivars of OSs. Therefore, we evaluated the metabolite compositions and AMPK activations in the seedlings of various oat cultivars at different growth times.

The purpose of this present work were to compare the policosanols and AMPK activation properties from OSs of Korean cultivars in growth times. Nine policosanols in the silylated hexane extracts of OSs were characterized by gas chromatography coupled with a single quadrupole mass spectrometry (GC-MS). We also evaluated potential cultivar and optimal conditions with high policosanol content to enhance the functional value of this species. In addition, our study was the first to document the degree of the viability and AMPK activation in HepG2 cells under abundant policosanols of OSs at different growth times.

## 2. Materials and Methods

### 2.1. Plant Material and Chemical Reagents

Fifteen Korean oat cultivars, namely, Gwanghan, Dahan, Donghan, Samhan, Shinhan, Okhan, Johan, Taehan, Punghan, Dakyung, Dajo, Jopung, Darkhorse, Hi-early, and High Speed were used in this research (Figure 1A). All cultivars of oat were planted in 2018 under artificial soil in a growth chamber from the National Institute of Crop Science (NICS), Rural Development Administration (RDA), Jeonbuk, Korea. Fifteen oat seeds were washed in water at 20 °C for 18 h, and then germinated at 65% humidity at 25 °C in the dark. The conditions were as follows: temperature, 20 °C; humidity, 60–70%; illumination intensity, 3300–5500 lx; light, 9 h → dark, 15 h (repeated alternatively). The OSs were cultured for 6 different growth times, counting the sowing day as 0 days as follows: 5 days (1st growth), 7 days (2nd growth), 9 days (3rd growth), 11 days (4th growth), 13 days (5th growth), and 15 days (6th growth) (Figure 1B). The harvested seedlings were washed with clean sterile water and freeze-dried at −78 °C. Hexane solvent and *N*-methyl-N-(trimethylysilyl) trifluoroacetamide (MSTFA) were obtained from Sigma-Aldrich (Sigma Co., St. Louis, MO, USA). Policosanol materials including eicosanol (PubChem CID: 12404), heneicosanol (PubChem CID: 85014), docosanol (PubChem CID: 12620), tricosanol (PubChem CID: 18431), tetracosanol (PubChem CID: 10472), hexacosanol (PubChem CID: 68171), heptacosanol (PubChem CID: 74822), octacosanol (PubChem CID: 68406), and triacontanol (PubChem CID: 68972) were also purchased from Sigma-Aldrich. Dulbecco’s modified eagle’s medium, fetal bovine serum, and antibiotics (streptomycin and penicillin) were acquired from Gibco BRL (Grand Island, NY, USA). Primary (anti-AMPK; antiphospho-AMPK) and secondary (anti-mouse IgG-HRP; anti-rabbit IgG-HRP) antibodies were provided by Santa Cruz Biotechnology Inc. (Dallas, TX, USA). Other solvents and chemical reagents were of analytical grade (Sigma-Aldrich).

### 2.2. Preparation of OSs and Policosanol Materials

The dried powdered OSs were extracted in a shaking incubator with hexane (10 mL) for 12 h at 25 °C. After centrifugation of the crude extract, the supernatant was filtered through a 0.45 μm syringe filter (Whatman Inc., Maidstone, UK). The hexane was removed using an evaporator under reduced pressured and resuspended in MSTFA (250 µL) and chloroform solution (0.5 mL) modified the methods of Choi [10] for the silylation reaction. To analysis of GC-MS, the silylated mixture was stirred with chloroform for 15 min at 60 °C. For quantitative analysis, the policosanol standards were also silylated with MSTFA under the same conditions. To quantify policosanol in OSs, a calibration curve was prepared using 5 concentrations (6.25, 12.5, 25 and 50 μg/mL) of each standard. The quantification of the calibration plot was evaluated using the peak areas of the policosanol standards, and the individual correlation coefficient (*r*^2^) was at least 0.998. The curve regression equation of policosanol and its coefficients are as follows. Eicosanol; *y* = 129,494 *x* +447,802, *r*^2^ = 0.998, Heneicosanol; *y* = 135,321 *x* + 314,175, *r*
^2^ = 0.998, Docosanol; *y* = 135,968 *x* − 262,734, *r*
^2^ = 0.999, Tricosanol; *y* = 133,917 *x* − 635,626, *r*
^2^ = 0.999, Tetracosanol; *y* = 124,434 *x* − 306,413, *r*
^2^ = 0.998, Hexacosanol; *y* = 144,421 *x* − 1,962,403, *r*
^2^ = 0.999, Heptacosanol; *y* = 135,627 *x* − 2,039,018, *r*
^2^ = 0.999, Octacosanol; *y* = 112,953 *x* − 841,945, *r*
^2^ = 0.998 and Triacntanol; *y* = 114,234 *x* − 2,274,380, *r*
^2^ = 0.999.

### 2.3. Instruments

The policosanol components were characterized and examined by an Agilent 7890A GC-MS (Agilent Technologies Inc., PaloAlto, CA, USA). The absorbance results were analyzed using a microplate plate reader (96 well, Molecular Devices, Sunnyvale, CA, USA). Lysate were transferred to nitrocellulose membranes by electrophoretic transfer cell (Bio-Rad, Hercules, CA, USA).

### 2.4. GC-MS Conditions for Policosanol Analysis

GC-MS analysis of policosanol was analyzed using previously published methods [12]. The silylated samples were examined by a GC system coupled with a 5977A series mass. Experimental conditions of GC-MS system were as follows: HP-5MS UI (diphenyl 5%-dimethylsiloxane 95% co-polymer) capillary column (30 m × 0.25 µm × 0.25 µm film thickness). The helium as the carrier gas with flow rate of at 1.2 mL/min. The ionization energy of MS spectrum was taken at 70 eV and mass range was 40–500 amu. The oven temperature was programmed to rise from 230 to 260 °C with 25 °C/min during 10 min, from 260 to 300 °C with 20 °C/min heating rate and maintained for 7 min. The temperatures of inlet and MS transfer line were 280 °C, and MS source was 230 °C, respectively. Samples were injected 1 µL by auto sampler using split mode injection (1:5). The policosanols were identified through the comparisons of fragmentation patterns of their mass values, retention times of the standards.

### 2.5. Cell Culture and Measurement of Cell Viability

HepG2 cell line was obtained from a Korean cell line bank (SPL Life Sciences, Pocheon, Korea). Cells were grown in Dulbecco’s modified Eagle’s medium (DMEM) supplemented with 10% fetal bovine serum (FBS), 1% glutamine, 1% antibiotics (penicillin-streptomycin), and 1.5 g/L sodium bicarbonate [15]. Cells were incubated at 37 °C, in humidified atmosphere containing 5% CO_2_, and all the experiments were performed in a clean atmosphere. Cell viability was determined by using the Thiazolyl blue tetrazolium bromide (MTT) method. The cell viability of the hexane extracts of oat seedlings and standard materials in HepG2 cell lines were assessed using the MTT assay as previously reported [12,15]. The HepG2 cells were seeded in 96 well plates at 1 × 10^4^/well and incubated with the different dose of the hexane extract of oat seedlings (concentrations 0, 12.5, 25, 50, 100, and 200 µg/mL) at 37 °C for 24 h in a humidified atmosphere (5% CO_2_). After the cells were washed with phosphate-buffered saline (PBS) treated with MTT solution (0.5 mg/mL) to the wells and incubated for 4 h. After incubation, cells were suspended in solubilization buffer of 50% dimethyl sulfoxide (DMSO). Subsequently, the absorbance was measured at 540 nm for absorbance value.

### 2.6. Western Blot Analysis

Immunoblotting was performed as described with slight modifications as Lee [23]. Brifely, HepG2 cells were lysed in lysis buffer (10 mM Tris-HCl (pH 7.5), 10 mM NaCl, 0.1 M EDTA, 0.1% NP40, 0.1% sodium deoxycholate, and 0.25 M sodium pyrophosphate) containing protease and phosphatase inhibitor cocktails (Sigma P2850) at 4 °C. Lysates of cell supernatants were separated on 10% acrylamide SDS-PAGE and then transferred to 0.2 µm nitrocellulose membranes using electrophoretic transfer cell. The combined membranes were blocked with Tris buffered saline (TBS) containing 5% non-fat milk at 25 °C for 60 min. The membranes were incubated with primary antibody, anti-AMPK, phosphor-AMPK, and anti-*β*-actin, at 4 °C for 12 h. After incubating with the primary antibody, the membrane is washed with TBS-T for three times. The secondary antibodies, anti-mouse and anti-rabbit immunoglobulin G, incubated with for 60 min at 25 °C and then washed with TBS-T again. The immunoreactive protein bands were detected with super signal picochemilumnescent stain and the protein concentration was determined using the Bradford assay with bovine serum albumin (BSA) as a standard [24]. The band intensities were visualised by a ChemiDoc XRS system (Bio-Rad) and quantified using Gel-Pro Analyser Software (Silk Scientific, Inc., Orem, UT, USA) [15].

### 2.7. Statistical Analysis

Each experiment of policosanol contents was performed three times, and all data are presented as the mean ± standard derivation (SD) and their differences in growth times were calculated by Duncan’s multiple range test by the statistical analysis software (SAS) 9.2 PC package (SAS Institute Inc. Cary, NC, USA). A *p*-value less than 0.05 was considered statistically significant.

## 3. Results and Discussion

### 3.1. Identification of Policosanol Compositions in OSs by GC-MS Analysis

In the current research, the policosanol contents in the OSs extracted with *n*-hexane, and analyzed using GC-MS. The policosanol structures and their representative chromatogram are shown in Figure 2. The retention times (*t_R_*) of individual policosanol was in the following order: peak 1 (*t_R_* = 3.45 min), 2 (*t_R_* = 4.09 min), 3 (*t_R_* = 4.75 min), 4 (*t_R_* = 5.73 min), 5 (*t_R_* = 7.05 min), 6 (*t_R_* = 10.63 min), 7 (*t_R_* = 12.51 min), 8 (*t_R_* = 13.60 min), and 9 (*t_R_* = 15.87 min). The mass spectra of the policosanol-Trimethylsilyl (TMS) compositions exhibited distinctive ion peaks of the [M + 15]^+^ fragmentation pattern owing to the loss of the methyl (CH_3_) moiety, facilitating the identification of individual policosanols. The [M + 15]^+^ ion peaks (*m*/*z*) of the individual authentic policosanol-TMS derivatives were 355.3, 369.4, 383.4, 397.4, 411.4, 439.4, 453.5, 467.5, and 495.5, respectively. Furthermore, their mass spectra exhibited the characteristic ions of TMS derivatives of primary alcohols because of fragment ions of C_4_H_9_^+^, OH-Si(CH_3_)^2+^, and CH_2_OSi-(CH3)^3+^ with *m*/*z* values of 57, 75, and 103, respectively. Therefore, nine peaks were identified by comparison of mass spectra and authentic standards. Eicosanol (**1**) (peak 1, C20-OH), heneicosanol (**2**) (peak 2, C21-OH), docosanol (**3**) (peak 3, C22-OH), tricosanol (**4**) (peak 4, C23-OH), tetracosanol (**5**) (peak 5, C24-OH), hexacosanol (**6**) (peak 6, C26-OH), heptacosanol (**7**) (peak 7, C27-OH), octacosanol (**8**) (peak 8, C28-OH), and triacontanol (**9**) (peak 9, C30-OH).

### 3.2. Changes in Policosanol Contents in the Seedlings of Oat Cultivars at Different Growth Times

Many studies have evaluated the policosanol contents in crop seedlings. Numerous researchers have also demonstrated that the metabolite contents showed remarkable differences according to the cultivars and growth periods [25,26]. Unfortunately, to the best of our knowledge, the policosanol contents in the seedlings of oat cultivars have not been investigated at different growth times. Therefore, we examined the policosanol contents of *n*-hexane extracts of OSs at six different growth times using GC-MS and their chromatograms are shown in Figure 3 (cv. Gwanghan). The fragment ions and mass data of policosanols are as follows: eicosanol (**1**) (peak 1, 370.3, 355.3, 103.0, 75.0, 55.1), heneicosanol (**2**) (peak 2, 384.4, 369.4, 103.0, 75.0, 55.1), docosanol (**3**) (peak 3, 398.4, 383.4, 103.0, 75.0, 55.1), tricosanol (**4**) (peak 4, 412.4, 397.4, 103.0, 75.0, 55.1), tetracosanol (**5**) (peak 5, 426.4, 411.4, 103.0, 75.0, 55.1), hexacosanol (**6**) (peak 6, 454.5, 439.4, 129.0, 97.1, 75.0, 55.1), heptacosanol (**7**) (peak 7, 468.5, 453.5, 103.0, 75.0, 55.1), octacosanol (**8**) (peak 8, 482.5, 467.5, 106.0, 75.0, 55.1), and triacontanol (**9**) (peak 9, 510.5, 495.5, 103.0, 75.0, 55.1). In addition, the individual and total policosanol contents of 15 different cultivars at six growth times (Sowing date is counted as 0 days. 5, 7, 9, 11, 13, and 15 days) are presented in Table 1 and their contents are expressed as mg/100 g of OSs. The total policosanols varied widely between cultivars and growth times as follows: 316.0–443.7 mg/100 g (5 days), 413.2–495.2 mg/100 g (7 days), 439.9–611.8 mg/100 g (9 days), 477.1–647.7 mg/100 g (11 days), 462.7–595.3 mg/100 g (13 days), and 367.8–569.8 mg/100 g (15 days), respectively. Especially, peak 6 (hexacosanol) exhibited the highest average values, with 337.8 (5 days, 88%), 416.8 (7 days, 89%), 458.9 (9 days, 90%), 490.0 (11 days, 91%), 479.2 (13 days, 90%), and 427.0 mg/100 g (15 days, 89%), respectively. The second main policosanol, octacosanol contents are as follows according to the growth times: [peak 8; 23.5 (5 days), 23.7 (7 days), 24.5 (9 days), 24.9 (11 days), 24.4 (13 days), and 24.0 mg/100 g (15 days)]. The average contents of other policosanols was as follows with the rank order of increase rates: tetracosanol (**5**) (18.2 mg/100 g) > docosanol (**3**) (7.5 mg/100 g), and the remaining compositions were not detected.

The growth times on 11 days through the seedlings of 15 oat cultivars exhibited the predominant average contents with 541.7 mg/100 g, followed by 13 days > 9 days > 15 days > 7 days with 532.2, 508.3, 479.5, and 463.8 mg/100 g, respectively, the lowest average contents were observed at 5 days (383.3 mg/100 g). In other words, the average policosanols increased 463.8 (7 days) → 508.3 (9 days) → 541.7 mg/100 g (11 days) after 7 → 11 days of growth. These observations can be primarily influenced by the most abundant hexacosanol (C26-OH, peak 6), this policosanol, which increased (average contents: 416.8; 7 days → 458.9; 9 days → 490.0 mg/100 g; 11 days) during these growth times. Our results support previous observations that the growth times of crops can strongly affect the policosanol contents. 26 Moreover, extending the growth times from 5 to 7 days considerably increased the total average policosanols in each cultivar (Table 1) and their contents showed 383.3 → 463.8 mg/100 g. In particular, the hexacosanol (**6**) content increased significantly with 337.8 (5 days) → 416.8 mg/100 g (7 days). Interestingly, when the oat seedlings are grown for longer times in 11 → 15 days, the average total policosanols decreased with 541.7 → 532.2 → 479.5 mg/100 g. The above findings have confirmed that the hexacosanol content (average: 490 → 479.2 → 427.0 mg/100 g) can be responsible for the main portion of total policosanols. Based on the considerations, the total policosanol contents in oat seedlings increased mainly during 11 days, while the remaining periods showed reduction phenomena. Therefore, our results suggest that the appropriate growth time regarding policosanol compositions in oat seedlings may be 11 days after sowing. Furthermore, we are confident that the environmental factors including the growth times may be affected with the policosanol contents [25]. The present data were similar to previously reported results concerning variations of phytochemicals at different harvest times [12]. In summarize, our data suggest that the total policosanol content of oat seedlings was closely related to hexacosanol (C-26), the major policosanol. Other components, C22-OH (docosanol peak 3), C24-OH (tetracosanol, peak 5), and C28-OH (octacosanol, peak 8) were observed low average contents with 7.5, 18.2, and 24.2 mg/100 g, respectively, at six different growth times of 15 cultivars. Although many researches have demonstrated that the metabolite contents increased during maturation of crops, our work showed that the policosanol contents were not dependent on the maturation times. Our previous study showed that wheat policosanol content may be affected by factors such as environmental factors and genotypes [27]. The current results support the development of suitable cultivars that can potentially be used as human health foods. Particularly, the oat cultivars such as Gwanghan and Dahan may be recommended as excellent sources owing to high policosanol contents of 640.1 and 647.7 mg/100 g on 11 days. To obtain more information concerning the beneficial effects in oat plant, our research was designed to document the comparison of policosanol in different organs (seeds, roots, and stems) of oat seedlings at 11 days growth times (Figure 4). Interestingly, the stem organ displayed high policosanol contents with 730.9 mg/100 g (docosanol: 70.6, tetracosanol: 70.1, hexacosanol: 515.8, and octacosanol: 74.4 mg/100 g), while other organs were not detected. Therefore, we confirmed that the stems of oat seedlings may be utilized as the best excellent source in terms of functional uses. The present work provided for the first time the policosanol contents in the seedlings of various oat cultivars through different growth times.

### 3.3. Properties of AMPK Phosphorylation in Oat Seedlings

AMPK regulates oxidative stress to control fat metabolism and the AMPK activation is best known to improve insulin resistance [12,15]. Previously, the policosanol components displayed considerable AMPK-activating abilities [15]. In addition, we demonstrated that the wheat and barley seedlings were detected potential properties of AMPK activation [26]. However, to the best of our knowledge, AMPK activation in oat seedlings has not been reported yet. Therefore, we investigated the activation of AMPK in the hexane extract of this crop. The OS extracts for AMPK phosphorylation in HepG2 cells were examined, and cell viability was determined using the MTT assay. To observe the changes in AMPK regulation, various concentrations (0, 12.5, 25, 20, 100, and 200 μg/mL, cv. Gwanghan) were used to measure cell viability. Cell viability was approximately 98% at a concentration of 200 μg/mL (Figure 5A). We also examined the relative effectiveness of each extract in different growth times. The extracts of oat seedlings of 9 and 11 days showed high ratios with 205 and 210% in comparison with positive control (100%) (Figure 5B). The previous study reported that hexacosanol was related to increased AMPK activation. 15 In this regard, we focused on the growth time of 11 days and evaluated the effects of different doses of the tested samples (25, 50, and 100 μg/mL) on rates of AMPK expression and phosphorylation in HepG2 cells using western blotting to stimulate AMPK phosphorylation by using a positive control (*β*-actin) (Figure 5C). As a result, the ratio of AMPK to phosphorylated AMPK increased in a dose-dependent manner. Our results indicate that the effects of OSs on AMPK activation can be correlated with high policosanol content. Consequently, the major policosanol of oat seedlings, hexacosanol (**6**) may be an excellent factor for AMPK activation. We believe that the oat seedlings on 11 days growth time may be considered a potential AMPK activation source for developing a new health functional food.

## 4. Conclusions

The present work was elucidated for the first time nine policosanol compositions by GC-MS analysis from the hexane extract of oat seedlings. We have also proven the variations of individual and total policosanols in Korean cultivars during different growth times of 5, 7, 9, 11, 13, and 15 days. Policosanols exhibited significant differences between cultivars and growth times, especially, hexacosanol (**6**) showed the predominant component with 88–91% of the total average content. In addition, this policosanol displayed remarkable variations according to the growth times as follows: 337.8 → 416.8 → 458.9 → 490.0 → 479.2 → 427.0 mg/100 g. Interestingly, the harvested seedlings on 11 days exhibited the highest average policosanols with 541.7 mg/100 g in 15 oat cultivars, and the remaining sources were in decreasing order: 532.2 (13 days) > 508.3 (9 days) > 479.5 (15 days) > 463.8 (7 days) > 383.3 mg/100 g (5 days). The hexane extract of 11 days was observed the highest effects of AMPK activation in HepG2 cells, and the main policosanol, hexacosanol (**6**) may be considered as excellent AMPK activator. Based on the above findings, the optimum growth time can be in 11 days due to the policosanol contents and AMPK activations, specifically, Gwanghan and Dahan cultivars may be considered as potential materials for functional agents using oat seedlings. We believe that the oat seedlings can be employed as an excellent material for improving human nutrition and health.

## Figures and Tables

**Figure 1 plants-11-01844-f001:**
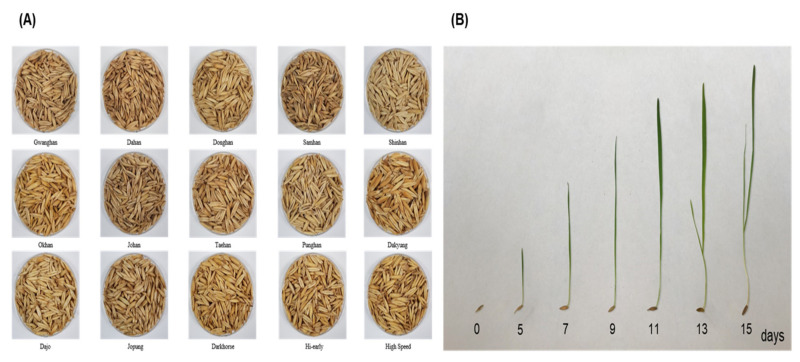
The appearances of oat seedlings (*Avena sativa* L.): (**A**) Oat seeds used in this study; (**B**) Oat seedlings (cv. Gwanghan) at six different growth times.

**Figure 2 plants-11-01844-f002:**
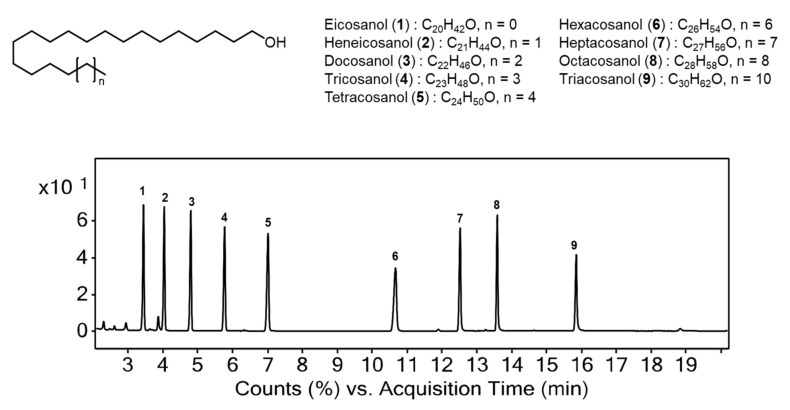
Chemical structures and GC-MS chromatogram of policosanol standards.

**Figure 3 plants-11-01844-f003:**
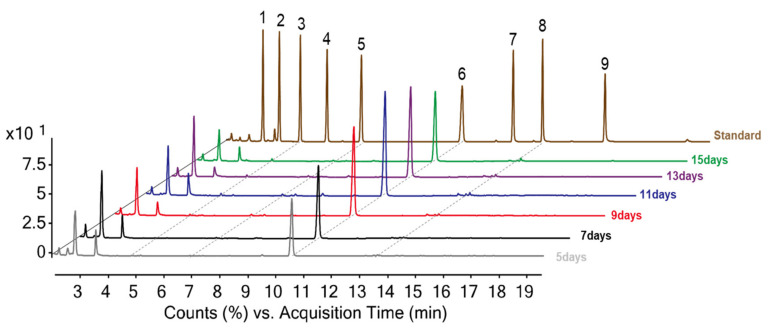
Comparison of GC-MS chromatograms for policosanol standard and oat seedlings at different growth times. (cv. Gwanghan). Eicosanol (**1**), Heneicosanol (**2**), Docosanol (**3**), Tricosanol (**4**), Tetracosanol (**5**), Hexacosanol (**6**), Heptacosanol (**7**), Octacosanol (**8**), Triacontanol (**9**).

**Figure 4 plants-11-01844-f004:**
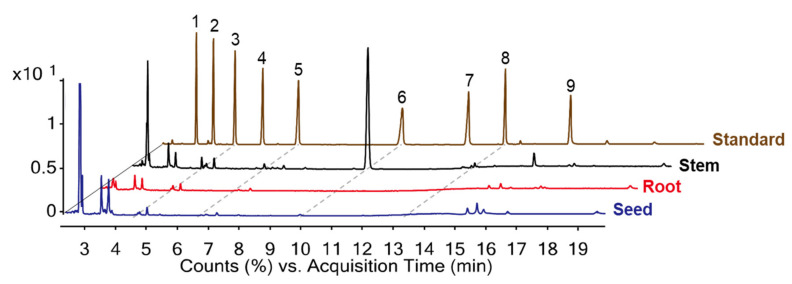
Policosanol in various parts of oat seedlings and standard GC-MS chromatogram (cv. Gwanghan). Eicosanol (**1**), Heneicosanol (**2**), Docosanol (**3**), Tricosanol (**4**), Tetracosanol (**5**), Hexacosanol (**6**), Heptacosanol (**7**), Octacosanol (**8**), Triacontanol (**9**).

**Figure 5 plants-11-01844-f005:**
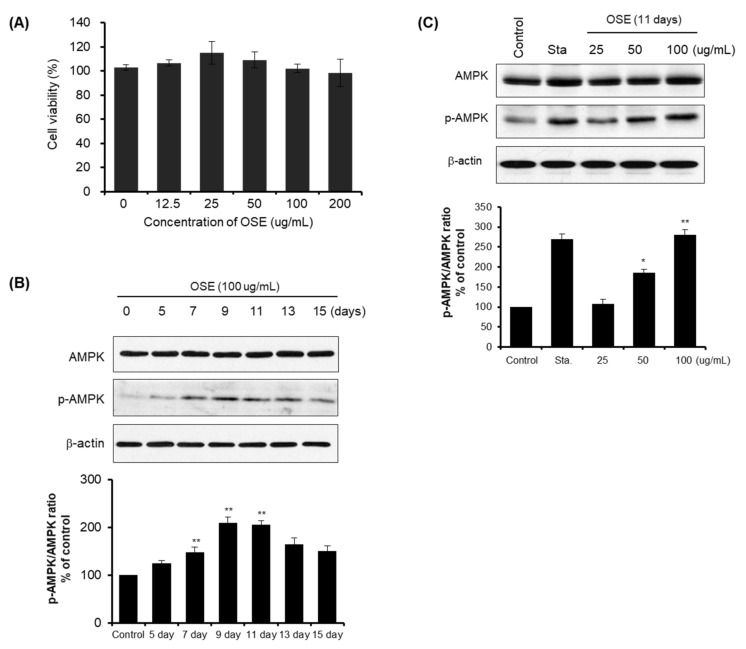
Effects of policosanol on HepG2 cells induced p-AMPK expression via activating AMPK. (**A**) Cell viability of each concentration (0, 12.5, 25, 50, 100, and 200 μg/mL) measured by MTT on HepG2 cells. (**B**) HepG2 cells were treated with OS extracts from different growth stages for different growth times (5, 7, 9, 11, 13, and 15 days) and p-AMPK expression was measured by western blot analysis. *β*-Actin was used as the internal standard. (**C**) HepG2 cells were treated with variable concentrations (25, 50, and 100 μg/mL) of OS extracts under 11 days. Under the same 11 days growth time, p-AMPK expression was measured by western blot analysis. Data represent the mean ± standard deviation of three independent experiments. (*) *p* < 0.05, (**) *p* < 0.01 when compared to corresponding control cells.

**Table 1 plants-11-01844-t001:** Changes of policosanol contents in the seedlings of oat cultivars at six different growth times.

PC Content (mg/100 g)											
Growth Time	Cultivar	C_20_	C_21_	C_22_	C_23_	C_24_	C_26_	C_27_	C_28_	C_30_	Total PC
5 days	Gwanghan	ND	ND	5.3 ± 0.0 ^cd^	ND	16.1 ± 0.0 ^fg^	339.7 ± 6.5 ^cde^	ND	23.1 ± 0.1 ^fgh^	ND	384.3 ± 3.2 ^cde^
	Dahan	ND	ND	5.3 ± 0.0 ^e^	ND	16.5 ± 0.1 ^de^	347.3 ± 4.9 ^bcd^	ND	23.3 ± 0.1 ^efg^	ND	392.4 ± 2.4 ^bcd^
	Donghan	ND	ND	5.3 ± 0.0 ^cd^	ND	16.3 ± 0.1 ^ef^	328.0 ± 12.2 ^def^	ND	23.0 ± 0.1 ^fg^	ND	372.6 ± 6.1 ^def^
	Samhan	ND	ND	5.3 ± 0.0 ^cde^	ND	16.0 ± 0.1 ^g^	322.5 ± 15.2 ^def^	ND	23.5 ± 0.1 ^cde^	ND	367.4 ± 7.5 ^def^
	Shinhan	ND	ND	5.3 ± 0.0 ^de^	ND	16.0 ± 0.1 ^g^	323.6 ± 18.6 ^def^	ND	23.5 ± 0.3 ^cde^	ND	368.4 ± 9.2 ^def^
	Okhan	ND	ND	5.3 ± 0.0 ^de^	ND	16.5 ± 0.1 ^d^	271.9 ± 5.4 ^g^	ND	22.9 ± 0.1 ^h^	ND	316.7 ± 2.7 ^g^
	Johan	ND	ND	5.3 ± 0.0 ^bc^	ND	17.5 ± 0.2 ^b^	364.5 ± 10.2 ^bc^	ND	24.0 ± 0.2 ^b^	ND	411.3 ± 5.0 ^bc^
	Taehan	ND	ND	5.4 ± 0.0 ^ab^	ND	17.3 ± 0.2 ^b^	371.7 ± 15.7 ^ab^	ND	23.9 ± 0.1 ^b^	ND	418.3 ± 7.8 ^ab^
	Punghan	ND	ND	5.3 ± 0.0 ^cde^	ND	16.7 ± 0.1 ^cd^	332.2 ± 10.2 ^def^	ND	23.3 ± 0.1 ^efg^	ND	377.6 ± 5.1 ^def^
	Dakyung	ND	ND	5.4 ± 0.0 ^a^	ND	17.8 ± 0.3 ^a^	396.6 ± 31.7 ^a^	ND	23.8 ± 0.3 ^bc^	ND	443.7 ± 15.7 ^a^
	Dajo	ND	ND	5.3 ± 0.0 ^cde^	ND	16.1 ± 0.2 ^fg^	315.7 ± 25. 1 ^ef^	ND	23.1 ± 0. 2 ^gh^	ND	360.3 ± 12.5 ^ef^
	Jopung	ND	ND	5.3 ± 0.0 ^bc^	ND	16.6 ± 0.1 ^d^	393.1 ± 11.1 ^a^	ND	24.3 ± 0.1 ^a^	ND	439.4 ± 5.5 ^a^
	Darkhorse	ND	ND	5.3 ± 0.0 ^ab^	ND	16.8 ± 0.2 ^c^	314.5 ± 20.6 ^ef^	ND	23.4 ± 0.2 ^def^	ND	360.2 ± 10.2 ^ef^
	Hi-early	ND	ND	5.3 ± 0.0 ^bc^	ND	16.3 ± 0.0 ^f^	304.7 ± 5.7 ^f^	ND	23.7 ± 0.0 ^bcd^	ND	350.0 ± 2.9 ^f^
	High Speed	ND	ND	5.3 ± 0.0 ^da^	ND	17.5 ± 0.1 ^b^	341.1 ± 9.1 ^bcd^	ND	23.7 ± 0.2 ^bcd^	ND	387.6 ± 4.5 ^cde^
7 days	Gwanghan	ND	ND	6.3 ± 0.0 ^c^	ND	17.4 ± 0.0 ^f^	435.3 ± 2.6 ^ab^	ND	23.7 ± 0.1 ^cde^	ND	482.7 ± 1.3 ^ab^
	Dahan	ND	ND	5.3 ± 0.0 ^h^	ND	16.9 ± 0.1 ^g^	439.6 ± 9.9 ^ab^	ND	23.8 ± 0.1 ^cde^	ND	485.5 ± 4.9 ^ab^
	Donghan	ND	ND	6.1 ± 0.0 ^d^	ND	17.5 ± 0.1 ^ef^	386.0 ± 4.7 ^cd^	ND	23.4 ± 0.1 ^e^	ND	433.0 ± 2.3 ^de^
	Samhan	ND	ND	5.4 ± 0.0 ^gh^	ND	16.8 ± 0.1 ^g^	439.3 ± 18.2 ^ab^	ND	24.3 ± 0.1 ^ab^	ND	485.8 ± 9.0 ^ab^
	Shinhan	ND	ND	5.3 ± 0.0 ^h^	ND	16.4 ± 0.1 ^h^	419.0 ± 13.8 ^abc^	ND	23.8 ± 0.2 ^cd^	ND	464.5 ± 6.9 ^abcd^
	Okhan	ND	ND	5.3 ± 0.0 ^h^	ND	17.4 ± 0.3 ^f^	392.6 ± 38.9 ^cd^	ND	23.5 ± 0.3 ^de^	ND	438.8 ± 19.3 ^cde^
	Johan	ND	ND	5.8 ± 0.0 ^e^	ND	18.1 ± 0.3 ^cd^	423.1 ± 24.4 ^abc^	ND	23.6 ± 0.2 ^de^	ND	470.7 ± 12.1 ^abcd^
	Taehan	ND	ND	5.8 ± 0.0 ^e^	ND	17.9 ± 0.2 ^de^	440.5 ± 18.8 ^ab^	ND	23.9 ± 0.2 ^bcd^	ND	488.0 ± 9.3 ^ab^
	Punghan	ND	ND	5.4 ± 0.0 ^h^	ND	16.8 ± 0.0 ^g^	368.1 ± 1.2 ^d^	ND	23.0 ± 0.0 ^f^	ND	413.2 ± 0.6 ^e^
	Dakyung	ND	ND	5.6 ± 0.0 ^f^	ND	18.4 ± 0.1 ^bc^	447.4 ± 8.4 ^a^	ND	23.7 ± 0.0 ^cde^	ND	495.2 ± 4.2 ^a^
	Dajo	ND	ND	5.5 ± 0.0 ^g^	ND	16.7 ± 0.2 ^gh^	387.1 ± 24.0 ^cd^	ND	23.6 ± 0.3 ^de^	ND	432.9 ± 11.9 ^de^
	Jopung	ND	ND	5.6 ± 0.0 ^f^	ND	16.9 ± 0.2 ^g^	433.8 ± 23.6 ^ab^	ND	24.4 ± 0.3 ^a^	ND	480.8 ± 11.7 ^abc^
	Darkhorse	ND	ND	7.0 ± 0.2 ^a^	ND	19.4 ± 0.6 ^a^	417.4 ± 38.4 ^abc^	ND	23.8 ± 0.4 ^cde^	ND	467.6 ± 19.0 ^abcd^
	Hi-early	ND	ND	5.4 ± 0.0 ^gh^	ND	16.8 ± 0.2 ^g^	404.2 ± 29.8 ^bcd^	ND	24.1 ± 0.3 ^abc^	ND	450.5 ± 14.8 ^bcde^
	High Speed	ND	ND	6.6 ± 0.1 ^b^	ND	18.8 ± 0.3 ^b^	418.5 ± 18.9 ^abc^	ND	23.9 ± 0.2 ^bcd^	ND	467.7 ± 9.3 ^abcd^
9 days	Gwanghan	ND	ND	8.6 ± 0.0 ^b^	ND	19.2 ± 0.1 ^c^	559.3 ± 10.9 ^a^	ND	24.7 ± 0.2 ^abc^	ND	611.8 ± 5.4 ^a^
	Dahan	ND	ND	5.4 ± 0.0 ^h^	ND	17.4 ± 0.0 ^gh^	537.0 ± 7.2 ^b^	ND	24.4 ± 0.1 ^de^	ND	584.1 ± 3.6 ^b^
	Donghan	ND	ND	6.7 ± 0.1 ^cd^	ND	18.3 ± 0.2 ^de^	495.0 ± 24.0 ^c^	ND	24.3 ± 0.2 ^de^	ND	544.4 ± 11.9 ^c^
	Samhan	ND	ND	5.5 ± 0.0 ^h^	ND	16.8 ± 0.0 ^i^	443.5 ± 2.8 ^def^	ND	24.3 ± 0.1^de^	ND	490.0 ± 1.4 ^ef^
	Shinhan	ND	ND	5.4 ± 0.0 ^h^	ND	16.5 ± 0.1 ^j^	463.4 ± 13.8 ^d^	ND	24.3 ± 0.1 ^de^	ND	509.7 ± 6.9 ^de^
	Okhan	ND	ND	5.5 ± 0.0 ^h^	ND	17.8 ± 0.1 ^f^	438.9 ± 5.5 ^ef^	ND	24.7 ± 0.0 ^bc^	ND	486.9 ± 2.7 ^ef^
	Johan	ND	ND	6.5 ± 0.0 ^e^	ND	18.2 ± 0.1 ^e^	391.6 ± 11.8 ^g^	ND	23.5 ± 0.1 ^f^	ND	439.9 ± 5.9 ^g^
	Taehan	ND	ND	6.9 ± 0.1 ^c^	ND	18.5 ± 0.1 ^d^	431.0 ± 11.9 ^f^	ND	24.7 ± 0.2 ^bc^	ND	481.0 ± 5.9 ^f^
	Punghan	ND	ND	5.5 ± 0.0 ^h^	ND	17.7 ± 0.1 ^f^	441.2 ± 9.8 ^def^	ND	24.1 ± 0.1 ^e^	ND	488.4 ± 4.9 ^f^
	Dakyung	ND	ND	6.3 ± 0.0 ^f^	ND	19.1 ± 0.1 ^c^	400.4 ± 2.9 ^g^	ND	24.1 ± 0.2 ^e^	ND	450.0 ± 1.4 ^g^
	Dajo	ND	ND	5.7 ± 0.0 ^g^	ND	17.6 ± 0.1 ^fg^	457.7 ± 10.1 ^de^	ND	24.4 ± 0.1 ^d^	ND	505.5 ± 5.0 ^de^
	Jopung	ND	ND	6.6 ± 0.0 ^de^	ND	17.5 ± 0.0 ^fg^	437.9 ± 5.7 ^ef^	ND	24.8 ± 0.1 ^ab^	ND	486.9 ± 2.8 ^ef^
	Darkhorse	ND	ND	12.2 ± 0.3 ^a^	ND	21.8 ± 0.3 ^a^	460.0 ± 10.2 ^de^	ND	24.5 ± 0.1 ^cd^	ND	518.5 ± 5.0 ^d^
	Hi-early	ND	ND	5.5 ± 0.0 ^h^	ND	17.3 ± 0.1 ^g^	462.5 ± 7.6 ^d^	ND	25.0 ± 0.1 a	ND	510.3 ± 3.8 ^de^
	High Speed	ND	ND	8.7 ± 0.2 ^b^	ND	20.2 ± 0.4 ^b^	463.6 ± 26.5^d^	ND	25.0 ± 0.3 ^a^	ND	517.4 ± 13.1 ^d^
11 days	Gwanghan	ND	ND	10.8 ± 0.7 ^c^	ND	19.9 ± 0.9 ^cd^	583.6 ± 73.9 ^a^	ND	25.7 ± 0.8 ^a^	ND	640.1 ± 36.5 ^a^
	Dahan	ND	ND	5.5 ± 0.0 ^j^	ND	18.0 ± 0.2 ^ef^	599.0 ± 30.9 ^a^	ND	25.2 ± 0.3 ^abc^	ND	647.7 ± 15.3 ^a^
	Donghan	ND	ND	8.0 ± 0.3 ^e^	ND	18.5 ± 0.5 ^e^	475.0 ± 53.4 ^bc^	ND	24.3 ± 0.5 ^de^	ND	525.8 ± 26.5 ^bc^
	Samhan	ND	ND	5.6 ± 0.0 ^ij^	ND	17.1 ± 0.1 ^g^	473.6 ± 8.3 ^bc^	ND	24.9 ± 0.1 ^bcd^	ND	521.2 ± 4.1 ^bc^
	Shinhan	ND	ND	5.7 ± 0.0 ^hij^	ND	16.6 ± 0.2 ^g^	431.1 ± 27.8 ^c^	ND	24.1 ± 0.3 ^e^	ND	477.5 ± 13.8 ^c^
	Okhan	ND	ND	6.0 ± 0.0 ^ghi^	ND	17.9 ± 0.1 ^ef^	429.1 ± 9.5 ^c^	ND	24.2 ± 0.1 ^de^	ND	477.1 ± 4.7 ^c^
	Johan	ND	ND	9.6 ± 0.1 ^d^	ND	19.7 ± 0.3 ^d^	471.7 ± 21.9 ^bc^	ND	24.1 ± 0.2 ^e^	ND	525.1 ± 10.8 ^bc^
	Taehan	ND	ND	8.1 ± 0.1 ^e^	ND	19.4 ± 0.0 ^d^	493.9 ± 1.6 ^bc^	ND	25.5 ± 0.0 ^ab^	ND	546.8 ± 0.8 ^bc^
	Punghan	ND	ND	5.8 ± 0.1 ^hij^	ND	18.3 ± 0.7 ^ef^	483.2 ± 75.6 ^bc^	ND	24.8 ± 0.8 ^bcde^	ND	532.1 ± 37.5 ^bc^
	Dakyung	ND	ND	8.2 ± 0.0 ^e^	ND	20.5 ± 0.1 ^bc^	479.9 ± 9.2 ^bc^	ND	25.1 ± 0.1 ^abc^	ND	533.8 ± 4.6 ^bc^
	Dajo	ND	ND	6.2 ± 0.1 ^g^	ND	18.0 ± 0.3 ^ef^	485.0 ± 33.3 ^bc^	ND	24.6 ± 0.4 ^cde^	ND	533.8 ± 16.5 ^bc^
	Jopung	ND	ND	7.6 ± 0.1 ^f^	ND	18.0 ± 0.2 ^ef^	496.2 ± 21.4 ^bc^	ND	25.4 ± 0.3 ^ab^	ND	520.2 ± 10.6 ^bc^
	Darkhorse	ND	ND	15.0 ± 0.1 ^a^	ND	23.3 ± 0.3 ^a^	514.4 ± 17.1 ^b^	ND	25.2 ± 0.2 ^abc^	ND	578.0 ± 8.4 ^b^
	Hi-early	ND	ND	6.1 ± 0.1 ^gh^	ND	17.8 ± 0.3 ^f^	485.8 ± 31.3 ^bc^	ND	24.7 ± 0.3 ^cde^	ND	524.4 ± 15.6 ^bc^
	High Speed	ND	ND	11.3 ± 0.2 ^b^	ND	20.9 ± 0.2 ^b^	485.0 ± 14.7 ^bc^	ND	25.2 ± 0.2 ^abc^	ND	542.4 ± 7.3 ^bc^
13 days	Gwanghan	ND	ND	10.4 ± 0.0 ^d^	ND	18.6 ± 0.1 ^e^	518.8 ± 26.7 ^ab^	ND	24.8 ± 0.2 ^abc^	ND	572.5 ± 13.3 ^ab^
	Dahan	ND	ND	5.9 ± 0.0 ^gh^	ND	17.8 ± 0.2 ^fghi^	547.1 ± 11.9 ^a^	ND	24.4 ± 0.3 ^bcde^	ND	595.3 ± 5.9 ^a^
	Donghan	ND	ND	11.6 ± 0.8 ^c^	ND	19.2 ± 0.7 ^d^	511.5 ± 60.0 ^abc^	ND	24.6 ± 0.6 ^abcd^	ND	566.9 ± 29.7 ^abc^
	Samhan	ND	ND	5.9 ± 0.0 ^h^	ND	17.5 ± 0.1 ^ghi^	487.5 ± 12.9 ^bcd^	ND	25.0 ± 0.1 ^a^	ND	535.8 ± 6.4 ^bc^
	Shinhan	ND	ND	6.6 ± 0.1 ^fg^	ND	17.3 ± 0.2 ^i^	484.6 ± 28.7 ^bcd^	ND	24.5 ± 0.2 ^abcde^	ND	533.0 ± 14.2 ^bc^
	Okhan	ND	ND	6.7 ± 0.1 ^f^	ND	18.1 ± 0.4 ^efg^	422.2 ± 31.0 ^bcd^	ND	23.7 ± 0.2 ^f^	ND	470.7 ± 15.4 ^de^
	Johan	ND	ND	17.0 ± 0.5 ^b^	ND	20.6 ± 0.3 ^c^	458.2 ± 16.7 ^ef^	ND	24.1 ± 0.2 ^ef^	ND	520.0 ± 8.2 ^bcd^
	Taehan	ND	ND	11.8 ± 0.1 ^c^	ND	19.7 ± 0.2 ^d^	463.4 ± 18.9 ^cdef^	ND	24.7 ± 0.2 ^abc^	ND	519.6 ± 9.3 ^bcd^
	Punghan	ND	ND	5.8 ± 0.0 ^h^	ND	18.3 ± 0.1 ^ef^	471.0 ± 10.8 ^bcde^	ND	24.0 ± 0.2 ^ef^	ND	519.2 ± 5.3 ^bcd^
	Dakyung	ND	ND	6.3 ± 0.1 ^fgh^	ND	19.2 ± 0.2 ^d^	463.1 ± 1.3 ^cdef^	ND	24.3 ± 0.1 ^cde^	ND	512.9 ± 0.6 ^cd^
	Dajo	ND	ND	6.1 ± 0.0 ^fgh^	ND	18.1 ± 0.1 ^efg^	478.5 ± 7.0 ^bcd^	ND	24.1 ± 0.1 ^def^	ND	526.8 ± 3.4 ^bc^
	Jopung	ND	ND	7.6 ± 0.1 ^e^	ND	18.0 ± 0.2 ^efgh^	511.7 ± 19.8 ^abc^	ND	24.8 ± 0.2 ^ab^	ND	562.1 ± 9.8 ^abc^
	Darkhorse	ND	ND	18.2 ± 0.9 ^a^	ND	23.1 ± 0.6 ^a^	490.7 ± 31.1 ^bcd^	ND	24.7 ± 0.4 ^abc^	ND	556.7 ± 15.2 ^abc^
	Hi-early	ND	ND	6.7 ± 0.1 ^f^	ND	17.4 ± 0.2 ^hi^	414.9 ± 32.2 ^f^	ND	23.7 ± 0.4 ^f^	ND	462.7 ± 16.0 ^e^
	High Speed	ND	ND	17.4 ± 0.8 ^b^	ND	22.2 ± 0.7 ^b^	465.2 ± 38.9 ^cdef^	ND	24.6 ± 0.3 ^abcd^	ND	529.4 ± 19.2 ^bc^
15 days	Gwanghan	ND	ND	11.9 ± 0.2 ^c^	ND	18.3 ± 0.1 ^cd^	406.0 ± 6.3 ^ef^	ND	23.4 ± 0.2 ^fgh^	ND	459.6 ± 3.1 ^de^
	Dahan	ND	ND	6.0 ± 0.0 ^h^	ND	17.0 ± 0.0 ^e^	436.2 ± 8.7 ^bcde^	ND	23.3 ± 0.1 ^ghi^	ND	482.5 ± 4.3 ^cde^
	Donghan	ND	ND	13.6 ± 0.3 ^b^	ND	18.6 ± 0.3 ^c^	414.6 ± 21.8 ^def^	ND	23.8 ± 0.5 ^defg^	ND	470.7 ± 10.7 ^cde^
	Samhan	ND	ND	6.4 ± 0.1 ^gh^	ND	18.2 ± 0.3 ^cd^	439.4 ± 23.4 ^bcde^	ND	24.7 ± 0.3 ^bc^	ND	488.7 ± 11.6 ^bcd^
	Shinhan	ND	ND	7.2 ± 0.1 ^ef^	ND	17.8 ± 0.1 ^d^	470.4 ± 14.9 ^abc^	ND	24.8 ± 0.2 ^b^	ND	520.1 ± 7.4 ^bc^
	Okhan	ND	ND	7.9 ± 0.2 ^d^	ND	18.3 ± 0.4 ^cd^	386.5 ± 37.0 ^f^	ND	23.5 ± 0.1 ^efgh^	ND	436.2 ± 18.4 ^e^
	Johan	ND	ND	15.4 ± 0.2 ^a^	ND	20.0 ± 0.1 ^b^	434.2 ± 8.8 ^bcde^	ND	23.8 ± 0.1 ^defg^	ND	493.4 ± 4.3 ^bcd^
	Taehan	ND	ND	13.5 ± 1.0 ^b^	ND	19.9 ± 0.9 ^b^	445.7 ± 58.5 ^bcde^	ND	24.8 ± 0.7 ^b^	ND	504.0 ± 28.8 ^bcd^
	Punghan	ND	ND	6.1 ± 0.1 ^h^	ND	18.4 ± 0.2 ^cd^	453.1 ± 20.2 ^bcde^	ND	24.0 ± 0.3 ^de^	ND	501.5 ± 10.0 ^bcd^
	Dakyung	ND	ND	6.7 ± 0.1 ^fg^	ND	19.5 ± 0.4 ^b^	460.1 ± 29.9 ^abcd^	ND	24.2 ± 0.2 ^cd^	ND	510.5 ± 14.8 ^bcd^
	Dajo	ND	ND	6.0 ± 0.1 ^h^	ND	16.6 ± 0.2 ^e^	322.3 ± 26.1 ^g^	ND	22.8 ± 0.2 ^i^	ND	367.8 ± 13.0 ^f^
	Jopung	ND	ND	7.5 ± 0.1 ^de^	ND	16.6 ± 0.1 ^e^	331.0 ± 19.1 ^g^	ND	23.1 ± 0.1 ^hi^	ND	378.3 ± 9.5 ^f^
	Darkhorse	ND	ND	15.8 ± 0.8 ^a^	ND	22.4 ± 0.6 ^a^	506.2 ± 30.9 ^a^	ND	25.4 ± 0.4 ^a^	ND	569.8 ± 15.2 ^a^
	Hi-early	ND	ND	7.4 ± 0.1 ^de^	ND	18.0 ± 0.2 ^d^	421.5 ± 14.1 ^cdef^	ND	24.0 ± 0.2 ^def^	ND	470.8 ± 7.0 ^cbd^
	High Speed	ND	ND	14.1 ± 0.2 ^b^	ND	21.9 ± 0.2 ^a^	477.4 ± 12.9 ^ab^	ND	24.8 ± 0.2 ^b^	ND	538.1 ± 6.4 ^ab^

All values are the mean ± SD derivation of three experiments. C_20_ = Eicosanol; C_21_ = Heneicosanol; C_22_ = Docosanol; C_23_ = Tricosanol; C_24_ = Tetracosanol; C_26_ = Hexacosanol; C_27_ = Heptacosanol; C_28_ = Octacosanol; C_30_ = Triacontanol; total PC, = Total policosanol content; ND = not detected; Data with different superscript letters differed significantly by date and cultivar with respect to each policosanol row. (Duncan’s multiple range test *p* < 0.05).

## Data Availability

The datasets used and/or analyzed during the current study are available from the corresponding author Woo-Duck Seo (swd2002@korea.kr) on reasonable request.

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
