# Peer review of "Comparative Analysis of Policosanols Related to Growth Times from the Seedlings of Various Korean Oat (Avena sativa L.) Cultivars and Screening for Adenosine 5′-Monophosphate-Activated Protein Kinase (AMPK) Activation"

_plants, 2022, doi:10.3390/plants11141844_

Round 1
Reviewer 1 Report
The study title Comparative analysis of policosanols related to growth times from the seedlings of various Korean oat (Avena sativa L.) cultivars and screening for adenosine 5´-monophosphate-activated protein knase (AMPK) activation made by Lee HG et al have very interesting results and is innovative.
In my opinion could be accepted for publishes after some minor improvements.
“Preparation of OSs and policosanol materials” must appear before the The points “2.2. Instruments” and “2.3. GC/MS conditions for policosanol analysis”
Line 224 beta must be in italic, revise in all document
Please be consistent use GC-MS or GC/MS
Figure 3 – identify what is the mean of STD
There are 2 Figures 3, Correct and correct please
Table 1 is not cited in the manuscript
In Table 1 the footnotes are impossible to read and what it means the “a” and “b” in the table titles. I think the authors can use these letters because they appear as result of Duncan’s multiple range test. The same for Table 2
In the tables the author must explain what does means the letters after the values and if it was the Duncan’s for each day or not
How the authors calculated the effect of the days? Explain better in the manuscript and in the results discussion.
Author Response
Dear Reviewer
Thank you very much for your kind consideration on our manuscript entitled “Comparative analysis of policosanols related to growth times from the seedlings of various Korean oat (Avena sativa L.) cultivars and screening for adenosine 5'-monophosphate-activated protein kinase (AMPK) activation”. We revised our manuscript according to the comments, and indicated the revised text with yellow color.
- Preparation of OSs and policosanol materials” must appear before the The points “2.2. Instruments” and “2.3. GC/MS conditions for policosanol analysis”.
Re: We revised it.
- Line 224 beta must be in italic, revise in all document.
Re: We revised it.
- Please be consistent use GC-MS or GC/MS.
Re: Thank you for the careful reading. We changed GC-MS in all documents.
- Figure 3 – identify what is the mean of STD.
Re: We modified figures and legends. Figure 4 has also been modified
- There are 2 Figures 3, Correct and correct please.
Re: We revised it.
- Table 1 is not cited in the manuscript.
Re: Thank you for the careful reading. We cited the Table 1 in line 276 and 306
.
- In Table 1 the footnotes are impossible to read and what it means the “a” and “b” in the table titles. I think the authors can use these letters because they appear as result of Duncan’s multiple range test. The same for Table 2.
Re: Footnotes are added with "a" to indicate how the statistical analysis was performed and "b" to indicate what the abbreviation means. However, since this can be interpreted as statistical significance, "a", "b", and "c" have been deleted and footnotes have been added to avoid confusion.
- In the tables the author must explain what does means the letters after the values and if it was the Duncan’s for each day or not
Re: We have attached note to the footnotes as follows
[ Data with different superscript letters differed significantly by date and cultivar with respect to each policosanol row. (Duncan’s multiple range test p <0.05) ]
- How the authors calculated the effect of the days? Explain better in the manuscript and in the results discussion.
Re: We calculated the sowing date as 0 days. We added it in the materials and methods and results. (Line 124 and 275)
thanks for the consideration
Your Sincerely​
Woo Duck Seo, Ph.D.
Researcher of Division of Crop Foundation,
National Institute of Crop Science (NICS),
Rural Development Administration (RDA)
181, Hyeoksin-ro, Iseo-myeon, Wanju-Gun, Jeollabuk-do, 55365, Korea
Tel: +82-63-238-5335, Fax: +82-63-238-5305
E-mail; [email protected]

Reviewer 2 Report
This is an interesting study; comments are as follows:
Line 117, what do the authors mean by ‘functional values’, pls be more specific.
Pls add reference to the method mentioned in 2.3 and 2.4.
Why do the authors select HepG2 cell line in this study?
Line 246-248, it’s common sense, can be deleted.
Line 255, What is TMS? Pls add the full name of the abbreviation(s) when mentioned for the first time. Pls double checked the whole manuscript and if there are more similar issues, pls revise them all.
The policosanols were tentatively identified by MS, however, it is better to do positive identification using authentic reference standards.
What is the quantification method for the nine policosanols? pls add detailed method and information to the manuscript.
Author Response
Dear Reviewer
Thank you very much for your kind consideration on our manuscript entitled “Comparative analysis of policosanols related to growth times from the seedlings of various Korean oat (Avena sativa L.) cultivars and screening for adenosine 5'-monophosphate-activated protein kinase (AMPK) activation”. We revised our manuscript according to the comments, and indicated the revised text with yellow color.
- Line 117, what do the authors mean by ‘functional values’, pls be more specific.
Re: According to your comments, we revised this sentence [ We also evaluated potential cultivar and optimal conditions with high policosanol content to enhance the functional value of this species. ] (Line 107-108)
- Pls add reference to the method mentioned in 2.3 and 2.4.
Re: Thank you for your comments. We added reference in 2.4
- Why do the authors select HepG2 cell line in this study?
Re: We selected HepG2 cells by focusing on the efficacy of inhibiting AMPK-mediated cholesterol synthesis in HepG2 cells, which is one of the representative effects of policosanol. And related sentences and references have been added. (Line 85-86)
- Line 246-248, it’s common sense, can be deleted.
Re: We deleted this sentence.
- Line 255, What is TMS? Pls add the full name of the abbreviation(s) when mentioned for the first time. Pls double checked the whole manuscript and if there are more similar issues, pls revise them all.
Re: We revised it.
- The policosanols were tentatively identified by MS, however, it is better to do positive identification using authentic reference standards.
Re: We confirmed the identification of policosanol of Oat seedling using GC-MS pattern and the retention time, comparing the standard form purchased from Sigma. This is mentioned in "Plant material and Chemical reagents" in Materials and methods and "Identification of policosanol compositions in OSs by GC-MS analysis" in Results and discussion
- What is the quantification method for the nine policosanols? pls add detailed method and information to the manuscript.
Re: Policosanol quantification was done through a calibration curve using the area value of the standard for each concentration. As pointed out, the quantitative method has been added in material and methods as follows. (Line 153-163)
[To quantify policosanol in OSs, a calibration curve was prepared using 5 concentrations (6.25, 12.5, 25 and 50 μg/ml) of each standard. The quantification of the calibration plot was evaluated using the peak areas of the policosanol standards, and the individual correlation coefficient (r2) was at least 0.998. The curve regression equation of policosanol and its coefficients are as follows. Eicosanol; y = 129494x+447802, r2 = 0.998, Heneicosanol; y = 135321x + 314175, r2 = 0.998, Docosanol; y = 135968 x − 262734, r2 = 0.999, Tricosanol; y = 133917 x – 635626, r2 = 0.999, Tetracosanol; y = 124434 x – 306413, r2 = 0.998, Hexacosanol; y = 144421 x – 1962403, r2 = 0.999, Heptacosanol; y = 135627 x – 2039018, r2 = 0.999, Octacosanol; y = 112953 x – 841945, r2 = 0.998 and Triacntanol; y = 114234 x – 2274380, r2 = 0.999 ].
Thanks for the consideration
Yours sincerely,
Woo Duck Seo, Ph.D.
Researcher of Division of Crop Foundation,
National Institute of Crop Science (NICS),
Rural Development Administration (RDA)
181, Hyeoksin-ro, Iseo-myeon, Wanju-Gun, Jeollabuk-do, 55365, Korea
Tel: +82-63-238-5335, Fax: +82-63-238-5305
E-mail; [email protected]

Round 2
Reviewer 2 Report
Further to my previous comment 6, in Line 288, authors said the 'nine peaks were tentatively identified as ....'. If the identification was confirmed with authentic standards, authors should state that the peaks were positively identified.
Author Response
Dear reviewer
We would like to thank reviewers for additional constructive and insightful suggestions, which helped us to improve our manuscript.
- Further to my previous comment 6, in Line 288, authors said the 'nine peaks were tentatively identified as ....'. If the identification was confirmed with authentic standards, authors should state that the peaks were positively identified.
RE: Thank you for pointing this out. We acknowledge the comment and we have modified the text.
[Therefore, nine peaks were identified by comparison of mass spectra and authentic standards. ]
Thanks for the consideration
Your Sincerely​
Woo Duck Seo, Ph.D.
Researcher of Division of Crop Foundation,
National Institute of Crop Science (NICS),
Rural Development Administration (RDA)
181, Hyeoksin-ro, Iseo-myeon, Wanju-Gun, Jeollabuk-do, 55365, Korea
Tel: +82-63-238-5335, Fax: +82-63-238-5305
E-mail; [email protected]